# Sensing Algorithm to Estimate Slight Displacement and Posture Change of Target from Monocular Images

**DOI:** 10.3390/s23020851

**Published:** 2023-01-11

**Authors:** Tadashi Ito, Hiroo Yoneyama, Yuto Akiyama, Tomonori Hagiwara, Shunsuke Ezawa

**Affiliations:** 1Graduate School of Science and Technology, Gunma University, Kiryu 376-8515, Gunma, Japan; 2Tokyo Measuring Instruments Laboratory Co., Ltd., Kiryu 376-0011, Gunma, Japan

**Keywords:** displacement sensor, 6DOF, monocular vision system

## Abstract

Various types of displacement sensors, which measure position changes of object, have been developed depending on the type and shape of the object under measurement, measurement range of the amount of displacement, required accuracy, and application. We are developing a new type of displacement sensor that is image-based, capable of measuring changes in 6DOF (3D position and orientation) of an object simultaneously, and is compact and low-cost. This displacement sensor measures the 6DOF of an object using images obtained by a monocular vision system. To confirm the usefulness of the proposed method, experimental measurements were conducted using a simple and inexpensive optical system. In this experiment, we were able to accurately measure changes of about 0.25 mm in displacement and 0.1 deg in inclination of the object at a distance of a few centimeters, and thus confirming the usefulness of the proposed method.

## 1. Introduction

Displacement sensors that measure changes in the position of an object are widely used in various fields, such as positioning of machine tools and semiconductor devices [1], vibration control of large structures [2], disaster prevention [3], structural health monitoring, etc. Various types of sensors have been developed according to the type and condition of the object under measurement, required range and accuracy of measurement, and response speed.

The methods of displacement sensors can be broadly classified into two types: contact and non-contact.

The contact type is that a sense terminal is in contact with the object. This type includes electric micrometers, which convert minute displacement of the terminal into an electrical quantity for measurement, and linear scales (also called linear encoders), which use light or magnetism to read a scale (scale position) on a straight line.

Contact sensors have the advantages of easy installation, high accuracy, and almost no influence from external disturbances, while they have the disadvantages of being limited to rigid objects, sometimes damaging objects, having a narrow measurement range, and being unsuitable for dynamic measurement while moving at high speed.

The non-contact type uses magnetism or lasers to measure without touching the object. This type includes optical sensors based on triangulation, linear encoders that optically or magnetically detect the displacement of a scale attached to an object, and laser length measuring machines based on Michelson-type interferometers.

Non-contact sensors have the advantages of not damaging objects, dynamic measurement, and a wide measurement range, but they also have the disadvantages of being affected by external disturbances, being restricted by the environment in which they are used, and requiring precision in installation and handling.

Most displacement sensors measure the distance in *z*-axis direction (out-of-plane displacement) based on the assumption that the object’s surface is directly facing the sensor, as shown in Figure 1a. On the other hand, sensors that can also measure displacement in the *x*-*y* plane of the object (in-plane displacement), as shown in Figure 1b, are almost exclusively image-based systems, as described below.

Information on the inclination of the object is also important, for example, in the measurement of slopes for the purpose of preventing landslides [4] and in the measurement of structures in structural health monitoring. As shown in Figure 2a, the inclination of an object is expressed in two angles: direction and degree. One method for estimating these angles is to calculate them from the displacement of three points forming a triangle on the surface of the object, but this method requires multiple sensors in case using contact-type displacement sensors.

When the displacement of an object is generalized to a 3D position and orientation, it is expressed in terms of six parameters: the amount of parallel displacement (position parameter), which is represented by the three components *x*, *y*, and *z*, and three angles (orientation parameter), which are the direction and degree of inclination, and rotation in the plane. They are called 6DOF parameters for short. There are few contact-type sensors that can measure 6DOF [5]. Most sensors that can measure 6DOF are image-based non-contact sensors [6].

The concept of image-based displacement sensor is shown in Figure 3. A pattern of known shape and precise dimension is placed on the target plane, and its image is captured by an image sensor (small camera) to measure the distance to the target plane (Method 1). Instead of setting a known pattern, feature points are extracted from the texture on the target plane, and the change of distance is estimated from the change in image at two different times (Method 2). This paper describes detailed algorithms for Method 1 and Method 2.

Image-based methods for measuring the 6DOF of an object can be further divided into two main categories, depending on whether the imaging system used is stereo vision or monocular vision.

In the method using stereo vision, the object is imaged by two cameras and the 6DOF is estimated based on the principle of binocular stereo vision [7].

The method using monocular vision, it estimates the position and the orientation of object from its images captured by a single camera [8,9,10]. In [8], a sampling moiré method is used to measure the displacement in the *z*-direction with 30 μm error for 20 cm distance, but it assumes that the displacement in the *x*-*y* direction is negligibly small. In [9], the displacement is estimated with an error of about 0.4 mm for a target with a distance of 1000 mm. However, the results of the posture measurements are not shown. In [10], 6DOF measurements were made on a ring-shaped object with a radius of 80 mm, with a displacement error of 0.05 mm and an angular error of 0.004 rad (0.23 deg) estimated at a distance of 40–50 cm. These studies evaluated errors in a few measurements and did not examine whether linearity was valid across the measurement range.

Our final goal is to develop a new compact and low-cost 6DOF displacement sensor. Image-based methods are the most promising for measuring 6DOF, and they also have the advantage of being non-contact. To meet the requirements of compactness and low cost, a monocular vision system with only one fixed camera is advantageous.

The measurement range and accuracy of our sensor are targeted to be equivalent to those of conventional contact-type displacement sensors, assuming that the new sensor may replace conventional sensors. Specifically, assuming a distance to the object of a few centimeters to a few dozen centimeters, the final target is a measurement accuracy of 2.5 µm error for a displacement measurement range of 5 mm, and an error of about 0.001 deg error for a tilt measurement range of ±1 deg. These measurement accuracy targets are high and challenging compared to previous studies, although the distances are closer.

When measuring displacement using images, displacement in the *z*-direction is in principle detected from the change in magnification of the object projected onto the image sensor. For an image point formed at a distance *x* from the optical axis on the image sensor, suppose that *x* changes by δx when the distance *z* changes slightly by δz. In this case, xz/f=(x+δx)(z−δz)/f holds where *f* is the focal length, from which δz≃z(δx/x) is obtained [11].

Based on the target specification, assuming that *z* = 20 cm, δz = 2.5 μm, and δx = 1 pixel is the limit of detection, then *x* = 80,000 pixels, indicating that an image sensor with very high resolution is required. Therefore, we first used an affordable image sensor to confirm the principle of the proposed method. 3280 × 2464 pixel image sensor with x=1000 pixel and z=25 cm would give δz=0.25 mm. Since the accuracy in the *z*-direction is 100 times greater than the final target, we set 0.1 deg as the target for this study, assuming that the accuracy in the angle measurement is similar.

This paper first explains the principle of measuring the 6DOF of an object using images obtained with a monocular vision system. To experimentally confirm the usefulness of the proposed method, we first conducted a experimental measurement using a simple and inexpensive optical system. In this experiment, although the final target measurement accuracy was not reached due to the limitation of the optical system, changes of about 0.25 mm in displacement and 0.1 deg in inclination could be measured with high accuracy, which we regard confirming the usefulness of the proposed method.

## 2. Related Works

The method of measuring the 6DOF of a target from an image has been the subject of much research in the field of computer vision because of its wide range of applications, including robot manipulation, SLAM (Simultaneous Localization and Mapping), and automated vehicle assistance. In these studies, the measurement items that are of importance depend on the application. For example, in robot manipulation, posture and accuracy in the *X* and *Y* directions are required to grasp the object, while the *Z* direction is relatively unimportant [12].

In contrast, obstacle detection in automatic driving does not require accuracy in orientation, but does require accuracy in the *Z* direction, which fluctuates greatly [13].

In [11], an error of 0.02 mm in the *Z* direction and an attitude angle of 15” at a distance of 10 m are achieved by supplementing the accuracy in the *Z* direction with a range finder, but the cost of the sensor increases.

In many studies, posture is estimated based on feature points extracted from images of the target object; in the case of 3D objects, the feature points that can be extracted vary depending on the viewing direction, and when multiple objects are present, feature point occlusion may occur. When extracting feature points from an image, anomalous points are often extracted. Therefore, assuming that a 3D shape model of the object is given in advance, the state-of-the-art issue is which of the extracted feature points should be selected and mapped to the model, and many methods using RANSAC [14] and deep learning have been developed.

Methods using deep learning require a large amount of training data, so methods using synthesized 3D data are also being studied [15]. For these method, changes in position and orientation between two points in time are learned from point cloud data. In [16], point clouds are extracted and mapped to obtain an accuracy of 0.816 degrees of rotation and 0.033 (relative value) of translation. Additionally, Ref. [17] also evaluates measurement accuracy using synthesized images and shows that it is possible to measure with an angular error of about 0.05 degrees. However, this was not evaluated using real images.

In contrast, in this study, the measurement target is always a single object, a surface of the object that is almost directly opposite to the sensor, and there is no problem of obscuring feature points or anomalous data. Correspondence of feature points extracted from images is straightforward because displacement and posture changes are slight. On the other hand, to the best of the author’s knowledge, no study has confirmed the accuracy within the measurement range for each of the 6DOF quantities from the images. Therefore, the contribution of this study is as follows.

The algorithm for estimating the 6DOF of a target based on a point cloud extracted from an image is presented in detail.The algorithm is based on solving a nonlinear optimization problem, and we show how to solve it using the L-BFGS method.The algorithm is applied to real images to measure the 6DOF, and the accuracy of each measurement quantity is examined separately.We will establish a method to measure 6DOF simultaneously and show the feasibility of a new 6DOF sensor using images.

## 3. Measuring Principle

First, the perspective projection transformation is described as an imaging model used in the proposed method. In this model, the object under measurement is represented as a point cloud, and the model shows the geometric relationship among the 3D coordinates of the relative positions of the point cloud, the 6DOF of the object, and the image coordinates of the point cloud in the captured image. Then, the algorithm for estimating the 6DOF of the object from the image coordinates of the point cloud will be presented.

In this study, we propose two measurement methods using images obtained by a monocular vision system.

In the first method (we call it Method 1), a pattern with a known arrangement of point clouds (a grid pattern is used in the experiment described below) is attached to an object, a group of points is extracted as feature points from a single image captured by a camera, then the 6DOF is estimated from the image coordinates. This method is known as a Perspective-n-Point problem in computer vision, and various solutions have been studied [18]. Most of these studies are based on parallel projection transform, which is linear and relatively easy to analyze as the imaging model. However, using this model makes it essentially impossible to obtain the distance to the object along the *z*-axis. Therefore, we use perspective projection transform, in which the problem leads to a multivariate nonlinear optimization and estimate numerically the 6DOF as the optimal solution of the problem.

In the second method (Method 2), in addition to 6DOF, the 3D coordinates of the relative arrangement of the point cloud of the object under measurement is also unknown to estimate. Each set of feature points corresponding to the point cloud is extracted from each of the images captured by a fixed camera at different time. After mapping the feature points corresponding between two images, the change in 6DOF between the two images will be estimated. This method is essentially equivalent to SfM (Structure from Motion) in computer vision [19], which simultaneously estimates the 6DOF of the camera and the 3D coordinates of the point cloud from multiple viewpoint images. To solve this problem, Tomasi–Kanade factorization method [20] and its extension to perspective projection transformation [21] are known. However, the main objective of SfM is to obtain the 3D shape of the object and does not focus on the measurement accuracy of 6DOF, while the main objective of our study is to obtain the 3DOF of the object with high accuracy, and the estimation algorithm has been improved.

### 3.1. Perspective Projection Transformation

As shown in Figure 4, set up an *X*-*Y*-*Z* axis (world coordinate system) with the center of the camera lens as the origin O and the optical axis as the *Z* axis (positive in the direction of the object being imaged). The *X* and *Y* axes should coincide with the column and row directions of the camera image, respectively. Note that this coordinate system is right-handed. Denote the focal length OC of the camera as *f*. Let (u,v) denote the position of a point on the image captured by the camera (image coordinates: unit is pixel) and a×a denote the pixel pitch. A point at position (X,Y,Z) in world coordinates is projected to position (u,v) in image coordinates. In the perspective projection model, the relationship between world coordinates and image coordinates is expressed as
(1)$$\left\{\begin{array}{cc} a(u-c_{u}) & =fX/Z\\ a(v-c_{v}) & =fY/Z \end{array}\right.$$
where $\left(c_{u},c_{v}\right)$ is the image coordinates of the point C (usually the center of the image) where the optical axis passes through the camera image. The above equation can be rewritten as
(2)$$\left\{\begin{array}{cc} u & =\kappa X/Z+c_{u}\\ v & =\kappa Y/Z+c_{v} \end{array}\right.$$
where κ=f/a.

Normally, internal parameters $\kappa ,c_{u},c_{v}$ of camera need to be experimentally determined using camera calibration. Currently, we simply used the catalog values of the camera used in the experiment (Raspberry Pi Camera Module V2 (Raspberry Pi Foundation, Cambridge, UK), Sony IMX219 image sensor (Sony Corporation, Tokyo, Japan), 3280×2464 pixels) with a=1.12
μm, f=3.04 mm, κ=3.04/1.12×103, and $\left(c_{u},c_{v}\right)=\left(1647.5,1255.5\right)$ (center of image).

A Cartesian coordinate system fixed to the object under measurement is called an object coordinate system. The basis vectors in the *X*-, *Y*-, and *Z*-axis of the object system are written as $X^{B},Y^{B},Z^{B}$. A position coordinate of a certain point is expressed as $r^{B}$ in the object system and $r^{I}$ in the world coordinate (Figure 5). The relationship between the two coordinates is expressed by the following equation.
(3)$$r^{I}=\left(X^{B}Y^{B}Z^{B}\right)r^{B}+r_{0}^{I}$$
where $r_{0}^{I}=\left(c_{x},c_{y},c_{z}\right)$ are the world coordinates of the object system origin, representing the translation of the object system. Additionally, $\left(X^{B}Y^{B}Z^{B}\right)$ is a rotation matrix with $X^{B}$, $Y^{B}$, and $Z^{B}$ as column vectors, representing the rotation of the object system. This rotation matrix is obtained by using the rotation matrix $R_{z}\left(\psi \right)$ of the angle ψ around the *Z* axis, the rotation matrix $R_{y}\left(\theta \right)$ of the angle θ around the *Y* axis, and the rotation matrix $R_{x}\left(\phi \right)$ of the angle ϕ around the *X* axis, and can be written as (rotated in the order of *Z*-axis, *Y*-axis, and *X*-axis)
(4)$$\left(X^{B}Y^{B}Z^{B}\right)=R_{z}\left(\psi \right)R_{y}\left(\theta \right)R_{x}\left(\phi \right)$$
where
(5)$$\begin{array}{cc} R_{x}\left(\phi \right) & =\left[\begin{array}{ccc} 1 & 0 & 0\\ 0 & \cos\phi & -\sin\phi \\ 0 & \sin\phi & \cos\phi \end{array}\right], \end{array}$$
(6)$$\begin{array}{cc} R_{y}\left(\theta \right) & =\left[\begin{array}{ccc} \cos\theta & 0 & \sin\theta \\ 0 & 1 & 0\\ -\sin\theta & 0 & \cos\theta \end{array}\right], \end{array}$$
(7)$$\begin{array}{cc} R_{z}\left(\psi \right) & =\left[\begin{array}{ccc} \cos\psi & -\sin\psi & 0\\ \sin\psi & \cos\psi & 0\\ 0 & 0 & 1 \end{array}\right]. \end{array}$$

### 3.2. 6DOF Estimation Algorithm

#### 3.2.1. Method 1: In Case of Imaging Pattern with Known Geometric Shape

The position coordinate of the point cloud in the object system is known, and expressed as $r_{i}={\left(x_{i},y_{i},z_{i}\right)}^{T}$(i=1,2,⋯,m), where *m* is the number of points and the superscript ${}^{T}$ represents the transpose of the matrix/vector. Let $p_{i}={\left(u_{i},v_{i}\right)}^{T}$ be the measured coordinates of the projection of point cloud onto the image. The coordinates ${\hat{p}}_{i}={\left({\hat{u}}_{i},{\hat{v}}_{i}\right)}^{T}$ on the image can be computed by the coordinate system rotation, translation and perspective projection model, and determined by ϕ, θ, ψ, $c_{x}$, $c_{y}$, $c_{z}$, $r_{i}$. Thus, it can be expressed in function p as ${\hat{p}}_{i}=p\left(\phi ,\theta ,\psi ,c_{x},c_{y},c_{z};r_{i}\right)$. The concrete computation procedure for the function p is expressed as follows:(8)$$\begin{array}{cc} \left(\begin{array}{c} X_{i}\\ Y_{i}\\ Z_{i} \end{array}\right) & =R_{z}\left(\psi \right)R_{y}\left(\theta \right)R_{x}\left(\phi \right)\left(\begin{array}{c} x_{i}\\ y_{i}\\ z_{i} \end{array}\right)+\left(\begin{array}{c} c_{x}\\ c_{y}\\ c_{z} \end{array}\right), \end{array}$$(9)$$\begin{array}{cc} \left(\begin{array}{c} {\hat{u}}_{i}\\ {\hat{v}}_{i} \end{array}\right) & =\frac{\kappa }{Z_{i}}\left(\begin{array}{c} X_{i}\\ Y_{i} \end{array}\right)+\left(\begin{array}{c} c_{u}\\ c_{v} \end{array}\right). \end{array}$$

Since the number of unknowns is six while the number of equations is 2m, it may be solved in case m≥3. In Method 1, the 3D coordinates $r_{i}$ of the point cloud and the image coordinates $p_{i}$ is given. Let $e_{i}$ residual
(10)$$\begin{array}{c} e_{i}={\hat{p}}_{i}-p_{i}=p\left(\phi ,\theta ,\psi ,c_{x},c_{y},c_{z};r_{i}\right)-p_{i} \end{array}$$
and *L* the square mean of the residuals
(11)$$\begin{array}{cc} L & =\frac{1}{2m}\sum_{i=1}^{m}\left|\right|e_{i}{\left|\right|}^{2}. \end{array}$$

Then the parameters phi, θ, ψ, $c_{x}$, $c_{y}$, and $c_{z}$ are estimated as the optimal solution that minimizes *L*. Since this minimization problem is nonlinear, iterative calculations are required to find the solution. Here, we use the limited-memory Broyden–Fletcher–Goldfarb–Shanno (L-BFGS) method, which is a type of quasi-Newton method and can be used for large-scale problems with high computational speed [22].

The calculation of the L-BFGS method requires computation of gradients of *L*, but not a Jacobian matrix. The gradient can be derived as follows. First, the partial derivative by ϕ is obtained by partial differentiation of Equation (Equation 11) as follows.
(12)$$\begin{array}{cc} \frac{\partial L}{\partial \phi } & =\frac{1}{m}\sum_{i=1}^{m}e_{i}^{T}\frac{\partial e_{i}}{\partial \phi } \end{array}$$
(13)$$\begin{array}{cc} \frac{\partial e_{i}}{\partial \phi } & =\frac{\partial {\hat{p}}_{i}}{\partial \phi }=-\frac{\kappa }{{Z_{i}}^{2}}\frac{\partial Z_{i}}{\partial \phi }\left(\begin{array}{c} X_{i}\\ Y_{i} \end{array}\right)+\frac{\kappa }{Z_{i}}\frac{\partial }{\partial \phi }\left(\begin{array}{c} X_{i}\\ Y_{i} \end{array}\right) \end{array}$$
(14)$$\begin{array}{cc} \frac{\partial }{\partial \phi }\left(\begin{array}{c} X_{i}\\ Y_{i}\\ Z_{i} \end{array}\right) & =R_{z}\left(\psi \right)R_{y}\left(\theta \right)\frac{\partial R_{x}(\phi )}{\partial \phi }\left(\begin{array}{c} x_{i}\\ y_{i}\\ z_{i} \end{array}\right) \end{array}$$

The same calculation is performed for partial derivation by θ and ψ. The partial derivatives by $c_{x}$ are as follows.
(15)$$\begin{array}{cc} \frac{\partial L}{\partial c_{x}} & =\frac{1}{m}\sum_{i=1}^{m}e_{i}^{T}\frac{\partial e_{i}}{\partial c_{x}} \end{array}$$
(16)$$\begin{array}{cc} \frac{\partial e_{i}}{\partial c_{x}} & =\frac{\partial {\hat{p}}_{i}}{\partial c_{x}}=-\frac{\kappa }{{Z_{i}}^{2}}\frac{\partial Z_{i}}{\partial c_{x}}\left(\begin{array}{c} X_{i}\\ Y_{i} \end{array}\right)+\frac{\kappa }{Z_{i}}\frac{\partial }{\partial c_{x}}\left(\begin{array}{c} X_{i}\\ Y_{i} \end{array}\right) \end{array}$$
(17)$$\begin{array}{cc} \frac{\partial }{\partial c_{x}}\left(\begin{array}{c} X_{i}\\ Y_{i}\\ Z_{i} \end{array}\right) & =\left(\begin{array}{c} 1\\ 0\\ 0 \end{array}\right) \end{array}$$
and thus the following equation is obtained.
(18)$$\frac{\partial {\hat{p}}_{i}}{\partial c_{x}}=\frac{\kappa }{Z_{i}}\left(\begin{array}{c} 1\\ 0 \end{array}\right)$$

Similarly for $c_{y},c_{z}$, the following equations are obtained.
(19)$$\frac{\partial {\hat{p}}_{i}}{\partial c_{y}}=\frac{\kappa }{Z_{i}}\left(\begin{array}{c} 0\\ 1 \end{array}\right),\;\frac{\partial {\hat{p}}_{i}}{\partial c_{z}}=-\frac{\kappa }{{Z_{i}}^{2}}\left(\begin{array}{c} X_{i}\\ Y_{i} \end{array}\right)$$

#### 3.2.2. Method 2: In Case of Imaging Pattern with Unknown Geometry

Coordinates of the object system $r_{i}$(i=1,2,⋯,m) are unknown and *n* images are taken while the position and orientation of the object changes. In this case, the following equation holds at j=1,2,⋯,n.
(20)$$\begin{array}{cc} \left(\begin{array}{c} X_{i,j}\\ Y_{i,j}\\ Z_{i,j} \end{array}\right) & =R_{z}\left(\psi_{j}\right)R_{y}\left(\theta_{j}\right)R_{x}\left(\phi_{j}\right)\left(\begin{array}{c} x_{i}\\ y_{i}\\ z_{i} \end{array}\right)+\left(\begin{array}{c} c_{x,j}\\ c_{y,j}\\ c_{z,j} \end{array}\right) \end{array}$$
(21)$$\begin{array}{cc} \left(\begin{array}{c} {\hat{u}}_{i,j}\\ {\hat{v}}_{i,j} \end{array}\right) & =\frac{\kappa }{Z_{i,j}}\left(\begin{array}{c} X_{i,j}\\ Y_{i,j} \end{array}\right)+\left(\begin{array}{c} c_{u}\\ c_{v} \end{array}\right) \end{array}$$

The first imaging (j=1) can be treated as the reference, in which the object system equals the reference system. Thus, $\phi_{1}$, $\theta_{1}$, $\psi_{1}$, $c_{x,1}$, $c_{y,1}$, $c_{z,1}$ are all zero and known.

Denoting the number of unknowns as *U* and the number of expressions as *C*, we have U=3m+6(n−1), C=2mn. In case n=1, we have U=3m, C=2m, which cannot be solved since U>C for any *m*. However, for n=2, U=3m+6, C=4m, so U≤C for m≥6 and may be solved.

As in Method 1, consider the problem of minimizing the mean of squares *L* of the residuals $e_{i,j}={\hat{p}}_{i,j}-p_{i,j}$
(22)$$\begin{array}{cc} L & =\frac{1}{2mn}\sum_{i=1}^{m}\sum_{j=1}^{n}\left|\right|e_{i,j}{\left|\right|}^{2} \end{array}$$

The gradient of *L* can be computed as follows (only the index *j* is included). First, the partial derivative by $\phi_{j}$ is as follows.
(23)$$\begin{array}{cc} \frac{\partial L}{\partial \phi_{j}} & =\frac{1}{mn}\sum_{i=1}^{m}e_{i,j}^{T}\frac{\partial e_{i,j}}{\partial \phi_{j}}\;\left(j=2,\ldots ,n\right) \end{array}$$
(24)$$\begin{array}{cc} \frac{\partial e_{i,j}}{\partial \phi_{j}} & =\frac{\partial {\hat{p}}_{i,j}}{\partial \phi_{j}}=-\frac{\kappa }{{Z_{i,j}}^{2}}\frac{\partial Z_{i,j}}{\partial \phi_{j}}\left(\begin{array}{c} X_{i,j}\\ Y_{i,j} \end{array}\right)+\frac{\kappa }{Z_{i,j}}\frac{\partial }{\partial \phi_{j}}\left(\begin{array}{c} X_{i,j}\\ Y_{i,j} \end{array}\right) \end{array}$$
(25)$$\begin{array}{cc} \frac{\partial }{\partial \phi_{j}}\left(\begin{array}{c} X_{i,j}\\ Y_{i,j}\\ Z_{i,j} \end{array}\right) & =R_{z}\left(\psi_{j}\right)R_{y}\left(\theta_{j}\right)\frac{\partial R_{x}(\phi_{j})}{\partial \phi_{j}}\left(\begin{array}{c} x_{i}\\ y_{i}\\ z_{i} \end{array}\right) \end{array}$$

The same calculation is performed for partial derivation by $\theta_{j}$, $\psi_{j}$. In addition, the partial derivative by $c_{x,j}$ is obtained as follows.
(26)$$\begin{array}{cc} \frac{\partial L}{\partial c_{x,j}} & =\frac{1}{mn}\sum_{i=1}^{m}e_{i,j}^{T}\frac{\partial e_{i,j}}{\partial c_{x,j}} \end{array}$$
(27)$$\begin{array}{cc} \frac{\partial e_{i,j}}{\partial c_{x,j}} & =\frac{\partial {\hat{p}}_{i,j}}{\partial c_{x,j}}=-\frac{\kappa }{{Z_{i,j}}^{2}}\frac{\partial Z_{i,j}}{\partial c_{x,j}}\left(\begin{array}{c} X_{i,j}\\ Y_{i,j} \end{array}\right)+\frac{\kappa }{Z_{i,j}}\frac{\partial }{\partial c_{x,j}}\left(\begin{array}{c} X_{i,j}\\ Y_{i,j} \end{array}\right) \end{array}$$
(28)$$\begin{array}{cc} \frac{\partial }{\partial c_{x,j}}\left(\begin{array}{c} X_{i,j}\\ Y_{i,j}\\ Z_{i,j} \end{array}\right) & =\left(\begin{array}{c} 1\\ 0\\ 0 \end{array}\right) \end{array}$$

From the last equation,
(29)$$\frac{\partial e_{i,j}}{\partial c_{x,j}}=\frac{\kappa }{Z_{i,j}}\left(\begin{array}{c} 1\\ 0 \end{array}\right)$$

The same calculation is performed for partial derivation of $c_{y,j}$, $c_{z,j}$, and the following equation is obtained.
(30)$$\frac{\partial e_{i,j}}{\partial c_{y,j}}=\frac{\kappa }{Z_{i,j}}\left(\begin{array}{c} 0\\ 1 \end{array}\right),\;\frac{\partial e_{i,j}}{\partial c_{z,j}}=-\frac{\kappa }{{Z_{i,j}}^{2}}\left(\begin{array}{c} X_{i,j}\\ Y_{i,j} \end{array}\right)$$

In Method 2, partial differentiation with respect to $x_{i}$ is also required, using the fact that the only terms involving $x_{i}$ are ${\hat{p}}_{i,j}$.
(31)$$\begin{array}{cc} \frac{\partial L}{\partial x_{i}} & =\sum_{j=1}^{n}e_{i,j}^{T}\frac{\partial e_{i,j}}{\partial x_{i}} \end{array}$$
(32)$$\begin{array}{cc} \frac{\partial e_{i,j}}{\partial x_{i}} & =\frac{\partial {\hat{p}}_{i,j}}{\partial x_{i}}=-\frac{\kappa }{{Z_{i,j}}^{2}}\frac{\partial Z_{i,j}}{\partial x_{i}}\left(\begin{array}{c} X_{i,j}\\ Y_{i,j} \end{array}\right)+\frac{\kappa }{Z_{i,j}}\frac{\partial }{\partial x_{i}}\left(\begin{array}{c} X_{i,j}\\ Y_{i,j} \end{array}\right) \end{array}$$
(33)$$\begin{array}{cc} \frac{\partial }{\partial x_{i}}\left(\begin{array}{c} X_{i,j}\\ Y_{i,j}\\ Z_{i,j} \end{array}\right) & =R_{z}\left(\psi_{j}\right)R_{y}\left(\theta_{j}\right)R_{x}\left(\phi_{j}\right)\left(\begin{array}{c} 1\\ 0\\ 0 \end{array}\right) \end{array}$$

#### 3.2.3. Improvement of Method 2

In the first stage of Method 2, the origin of the world coordinate system is used as the center point of the posture change. Therefore, if the posture changes slightly, the angular change when centered at the origin is minuscule, making estimation difficult. Additionally, the convergence of the iterative calculation becomes slow.

To solve this problem, we modified Method 2 so that the origin of the posture change is the center of gravity of the point cloud *g*.
(34)$$\begin{array}{cc} g & =\frac{1}{m}\sum_{i=1}^{m}r_{i} \end{array}$$
(35)$$\begin{array}{cc} \left(\begin{array}{c} X_{i,j}\\ Y_{i,j}\\ Z_{i,j} \end{array}\right) & =R_{z}\left(\psi_{j}\right)R_{y}\left(\theta_{j}\right)R_{x}\left(\phi_{j}\right)\left(r_{i}-g\right)+g+\left(\begin{array}{c} c_{x,j}\\ c_{y,j}\\ c_{z,j} \end{array}\right) \end{array}$$

Note that the calculation of the center of gravity also includes unknowns. The calculation of the gradient in Method 2 changes only in the following parts.
(36)$$\begin{array}{cc} \frac{\partial }{\partial \phi_{j}}\left(\begin{array}{c} X_{i,j}\\ Y_{i,j}\\ Z_{i,j} \end{array}\right) & =R_{z}\left(\psi_{j}\right)R_{y}\left(\theta_{j}\right)\frac{\partial R_{x}(\phi_{j})}{\partial \phi_{j}}\left(r_{i}-g\right) \end{array}$$
(37)$$\begin{array}{cc} \frac{\partial }{\partial x_{i}}\left(\begin{array}{c} X_{i,j}\\ Y_{i,j}\\ Z_{i,j} \end{array}\right) & =R_{z}\left(\psi_{j}\right)R_{y}\left(\theta_{j}\right)R_{x}\left(\phi_{j}\right)\left(\begin{array}{c} 1-1/m\\ 0\\ 0 \end{array}\right)+\left(\begin{array}{c} 1/m\\ 0\\ 0 \end{array}\right) \end{array}$$

## 4. Experiment

The proposed methods utilize images to measure the displacement, and orientation of the object. To obtain these information from images, we search for feature points of the object in the image and use their image coordinates.

### 4.1. Experimental Equipment

Figure 6 shows the appearance of the experimental setup. A grid pattern (CBBG01-150T manufactured by Shibuya Optical, Wako-shi, Japan) was used as the object under measurement. This is a quartz glass plate with a square grid of 30 × 30 squares with a grid spacing of 5 mm.

In this experiment, Raspberry Pi Camera Module V2 was used. A *z*-axis stage (Chuo Precision Industrial LV-6042-8; Tokyo, Japan) and a tilt stage (TS-613) were used to capture images by applying minute and precise displacement and tilt to the object. The resolution of the setting is 0.22 mm on a single scale for the *z*-axis stage and approximately 1°1′22″ on a single knob turn for the tilt stage.

The grid patter was placed at a distance of approximately 70 mm from the camera, almost directly opposite each other. Precise displacements of −2, −1, −0.5, −0.25, −0.1, +0.1, +0.25, +0.5, +1, +2 mm in the *Z* axis direction was applied by using the *z*-axis stage. For each image, the grid points of the grid pattern were detected, and the displacement was estimated by using Methods 1 and 2.

Among the detected feature points in the obtained images, 9 × 9 grid points near the center of the image were used for estimation, to avoid image distortion at the edges of the image.

### 4.2. Feature Extraction

Harris corner detection in OpenCV (version 2.4.9.1) was used to detect grid points. The principle of the algorithm is briefly described below. To detect corners, the first step is to find the difference in pixel values for a given pixel position (u,v) shift in all directions. This can be expressed by the following equation.
(38)$$\begin{array}{cc} E(u,v) & =\sum_{x,y}w\left(x,y\right){\left[I\left(x+u,y+v\right)-I\left(x,y\right)\right]}^{2} \end{array}$$
where w(x,y) is a window function representing the weight on each pixel.

For corner detection, the coordinates (u,v) that maximize E(u,v) are obtained. Specifically, maximize I(x,y) in Equation (Equation 38). The following equation is derived from Equation (Equation 38).
(39)$$\begin{array}{cc} E(u,v) & ≃\left[uv\right]M\left[\begin{array}{c} u\\ v \end{array}\right] \end{array}$$
where *M* is a matrix defined as follows.
(40)$$\begin{array}{cc} M & =\sum_{x,y}w\left(x,y\right)\left[\begin{array}{cc} I_{x}I_{x} & I_{x}I_{y}\\ I_{x}I_{y} & I_{y}I_{y} \end{array}\right] \end{array}$$

In addition, $I_{x}$ and $I_{y}$ represent the gradient in the *x* and *y* directions, respectively, and are computed with Sobel filters using the following weighting factors.
(41)$$\begin{array}{cc} I_{x} & :\left[\begin{array}{ccc} -1 & 0 & 1\\ -2 & 0 & 2\\ -1 & 0 & 1 \end{array}\right],\;I_{y}:\left[\begin{array}{ccc} -1 & -2 & -1\\ 0 & 0 & 0\\ 1 & 2 & 1 \end{array}\right] \end{array}$$

Note that these are the weight coefficients for an 8-neighborhood case. Sobel filters with 11×11 size weight coefficients were used in this study.

After the above process, the score *R* defined by the following equation is calculated to determine whether the corners are included in the search window.
(42)$$\begin{array}{cc} R & =\det\left(M\right)-k{\left(\mathrm{trace}\left(M\right)\right)}^{2} \end{array}$$
where $\det\left(M\right)=\lambda_{1}\lambda_{2}$ and $\mathrm{trace}\left(M\right)=\lambda_{1}+\lambda_{2}$, $\lambda_{1}$ and $\lambda_{2}$ are eigenvalues of *M*.

These eigenvalues determine whether the region of interest is a corner, an edge, or a flat region; if *R* is large, i.e., both $\lambda_{1}$ and $\lambda_{2}$ are large, the region is considered a corner. In this study, taking into account the performance of the optical system, image information other than the grid pattern due to the shooting environment, and the line thickness of the grid pattern, the arguments of the OpenCV cornerHarris function were set to blocksize = 10, *k*size = 11, and *k* = 0.08. Here, blocksize is the size of the adjacent regions considered in corner detection, ksize is the kernel size of the gradient operator given to the Sobel function that calculates the gradients $I_{x}$ and $I_{y}$, and *k* is the thresholds for corner detection. Since the size of the adjacent region considered for corner detection is set to 10, multiple points per grid point are detected as corners when detecting grid points. Therefore, in this study, the average pixel coordinates of a group of pixels close to a grid point are calculated, rounded to the nearest whole number, and used as the pixel coordinates of the grid point.

### 4.3. Experimental Results

Figure 7 and Figure 8 show the results of distance estimation by Methods 1 and 2, respectively. The horizontal and vertical axis represent the true and the estimated displacement along the *z*-axis, respectively. Straight lines were determined by applying the least-squares method to the data.

For Method 1, the horizontal axis represents the displacement in the *z*-axis from the reference position, whereas the vertical axis directly represents the distance from the camera to the grid pattern. Although it is difficult to determine the true value, the estimated distances show reasonable values.

For Method 2, both the horizontal and vertical axes represent the displacement in the *z*-axis direction between the two images acquired, thus the origin indicates that both the true and estimated values are undisplaced. The fitted line should pass through the origin, and indeed it does.

For both Methods 1 and 2, the data fit the straight line well and the slope is close to 1, indicating that the estimation is accurate.

Next, Figure 9 and Figure 10 show the results of the angle estimation for each methods when a change in inclination is given. In the figures, ’theta’ indicates the angle of inclination θ.

One possible cause of this is the effect of geometric distortion of the imaging system. When tilted, the grid pattern image is deformed from the orthogonal grid, and the degree of deformation represents the tilt. Therefore, the effect of image distortion is likely to appear. If image distortion is reduced by camera calibration, the slope of the fitted line will also approach 1.

Figure 11 and Figure 12 show the change in the estimated values of the iterations for Methods 1 and 2, respectively. Each of the plots represents the results for the different displacements ±1.0 and 0.0 mm, respectively.

Method 1 converges quickly to a constant value after about 30 iterations because it uses accurate information on the geometry of the object being measured. On the other hand, Method 2 does not use shape information, thus convergence is slow, requiring about 600 iterations. In any case, it can be seen that the convergence to a constant value is achieved after repeated iterations.

Table 1 shows the summary of the experimental results obtained by fitting the straight line y=ax+b to the true value *x* and the measured value *y*, measured with the proposed methods by varying *x*, *y*, *z*, ψ and θ among 6DOF. For ϕ, experiments are in progress. The closer the slope *a* of the straight line is to 1 and the closer the coefficient of determination $r^{2}$ is to 1, the higher the measurement accuracy. For Method 2, the closer the intercept *b* is to 0, the higher the measurement accuracy.

The table of experimental results shows that the coefficients of determination $r^{2}$ are all close to 1, indicating that a linear relationship is well-established between the estimated and true values. For Method 1, the slope of the line *a* is also close to 1. In contrast, the slope of Method 2 is slightly different from 1. The reason is considered to be that Method 1 uses a precise shape pattern of the object to be measured, whereas Method 2 does not, and thus the distortion of the imaging system has a significant effect on the results.

Comparing the maximum error of the experimental results with the targeted values (0.25 mm for displacement and 0.1 deg for angle), the target is achieved except for the last item (measurement of tilt angle θ using Method 2). Additionally, the measurement of the tilt angle by Method 2 is only slightly worse than the target accuracy.

Since no camera calibration has been applied in the above experiments, the re-projection errors are not very small due to the effect of lens distortion in the acquired images. Therefore, it is expected that camera calibration will improve the measurement accuracy.

To confirm this expectation, we performed a camera calibration and compared the measurement accuracy before and after calibration. OpenCV was used for camera calibration, and a checkerboard pattern was used instead of a grid pattern as the target, which was then used as the object under measurement.

Figure 13 and Figure 14 show the results. After camera calibration, the slope *a* was close to 1, confirming the effectiveness of the calibration. The maximum errors were 0.0693 mm and 0.0905 mm, respectively. In both cases, the targets were achieved. The reason that the maximum error became smaller even without camera calibration is considered to be that the positional accuracy of feature point extraction becomes higher when the measurement target is changed to a checkerboard pattern.

## 5. Discussion

The results of the displacement along the *z*-axis were close to the actual displacement and agree with the estimated distance. In addition, displacements as small as 0.1 mm could be measured.

For angle measurement, the results are a little less accurate than the change in the *z*-axis. This is due to the fact that optical distortion correction and optical axis alignment were not performed. Although the experimental results are still limited to a few measurement items, they show that camera calibration improves measurement accuracy, especially the slope of the linear relationship between the true value and the estimated value approaches well to 1. In terms of accuracy, the results of this experiment are better than the initially set goal.

## 6. Conclusions

In this study, we proposed an algorithm to estimate the minute displacement and inclination of an object using images, and developed a prototype system to measure the minute displacement. As a result, highly accurate results were obtained for the measurement of displacement. The measurement experiment after camera calibration confirmed that the initial target values of measurement error of 0.25 mm or less for displacement and 0.1 deg or less for angle were achieved for the 6DOF measurement items except for the angle ϕ.

Experiments are currently underway to confirm the measurement accuracy of the other parameters of 6DOF.

Further, designing a dedicated optical system for higher resolution and accuracy, implementing as a sensor device, and conducting experimental measurement under various type of objects and environments are subjects for future study.

## Figures and Tables

**Figure 1 sensors-23-00851-f001:**
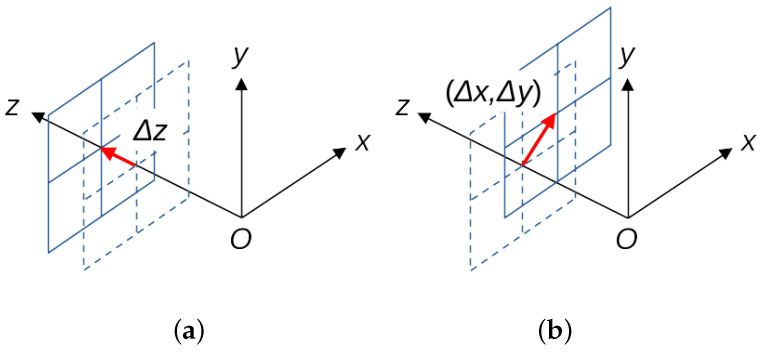
(**a**) Displacement in the direction of *z*-axis, and (**b**) 2D displacement in *x*-*y* plane.

**Figure 2 sensors-23-00851-f002:**
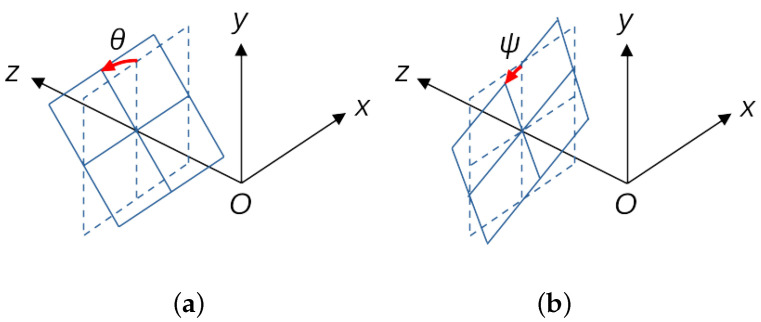
(**a**) Inclination of the object surface due to rotation around x-axis. The direction of the rotation axis is also one of the degrees of freedom. (**b**) In-plane rotation.

**Figure 3 sensors-23-00851-f003:**
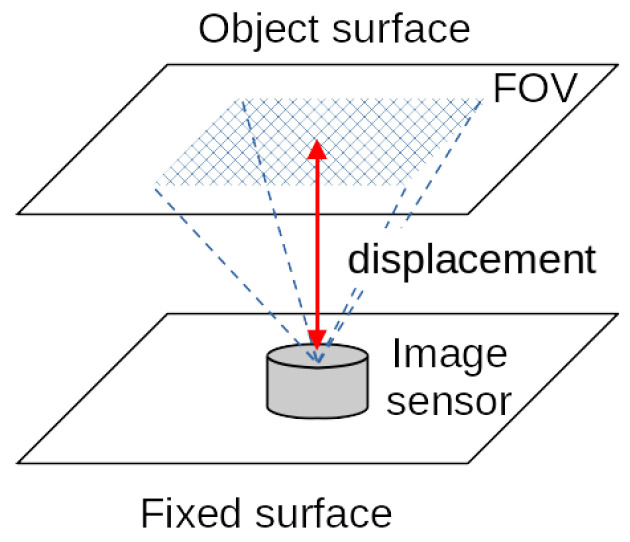
Conceptual diagram of an image-based displacement sensor. For example, it is installed in the gap between two structures and monitor changes in the distance between them. An image sensor is fixed on one side (fixed plane), and an image of the other side (object plane) is acquired to measure the displacement.

**Figure 4 sensors-23-00851-f004:**
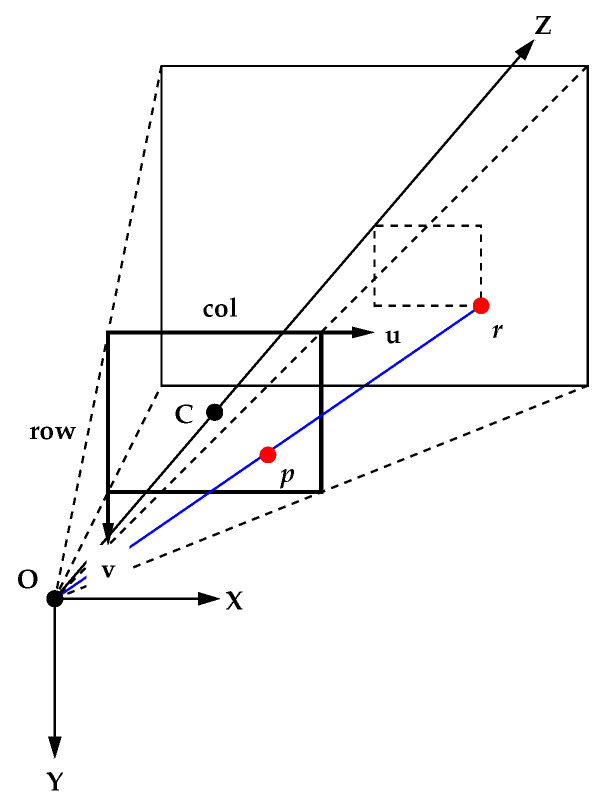
Perspective projection transformation.

**Figure 5 sensors-23-00851-f005:**
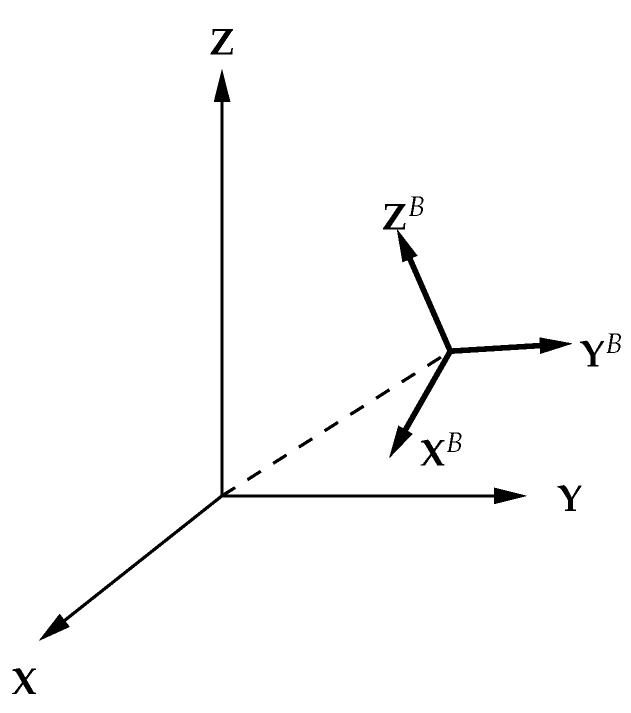
Rotation and translation of coordinate systems.

**Figure 6 sensors-23-00851-f006:**
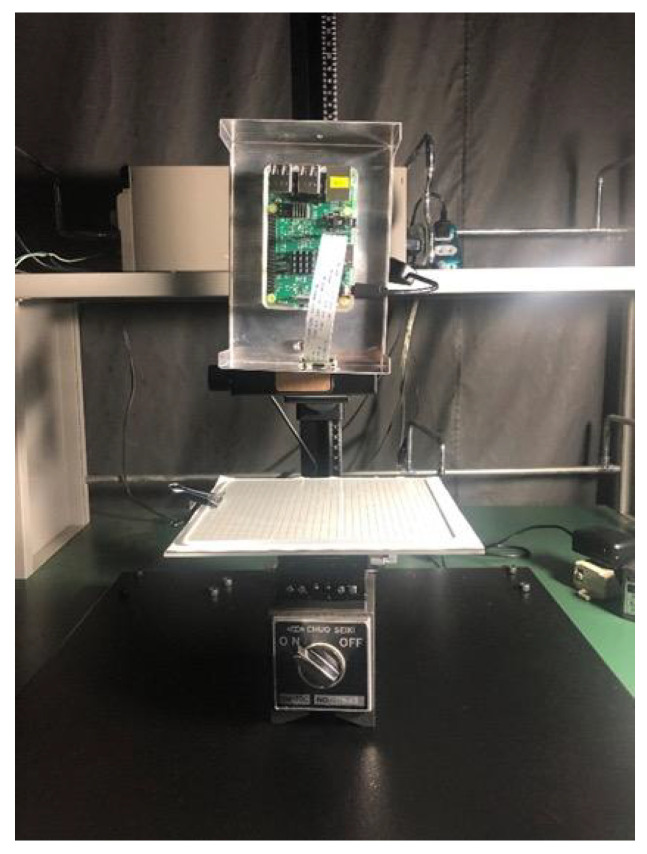
Appearance of the experimental setup. An optical stage is fixed on an optical surface plate, and a glass grid pattern placed on the optical stage is used as the object under measurement. The camera is fixed above the grid pattern. Images are acquired while the grid pattern is precisely displaced by the optical stage.

**Figure 7 sensors-23-00851-f007:**
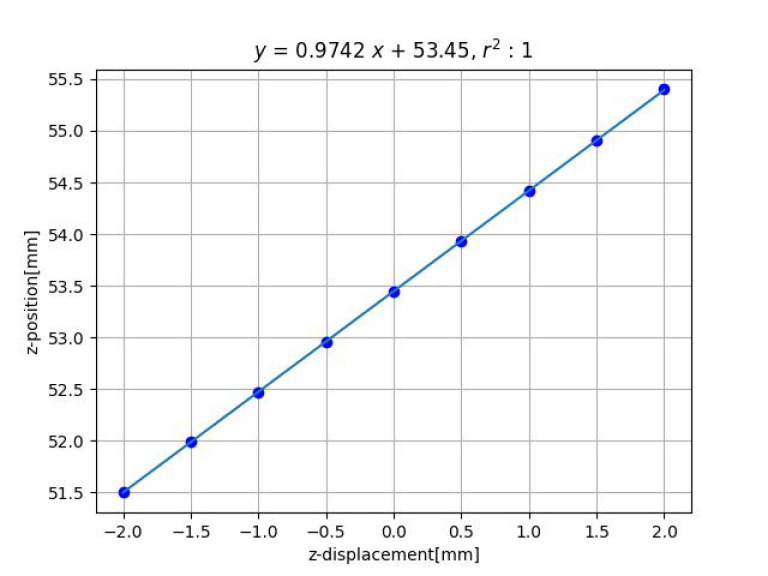
Result of estimation for *z*-axis displacement by method 1.

**Figure 8 sensors-23-00851-f008:**
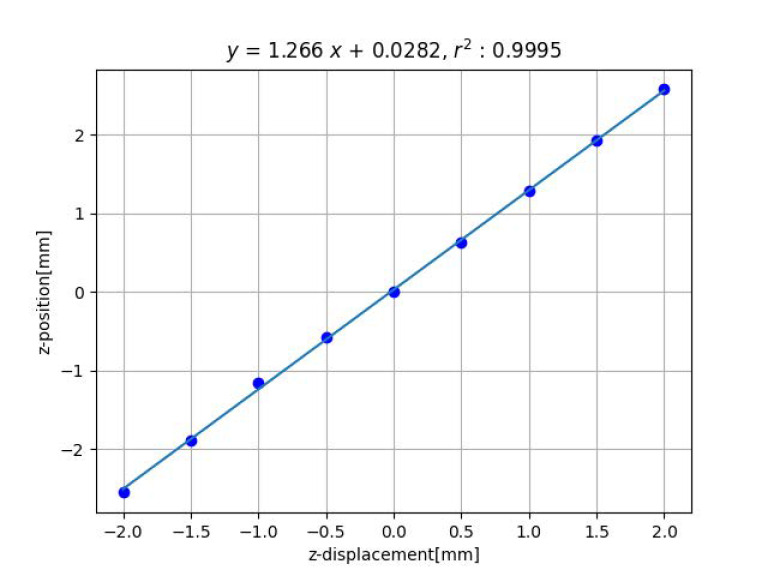
Result of estimation for *z*-axis displacement by method 2.

**Figure 9 sensors-23-00851-f009:**
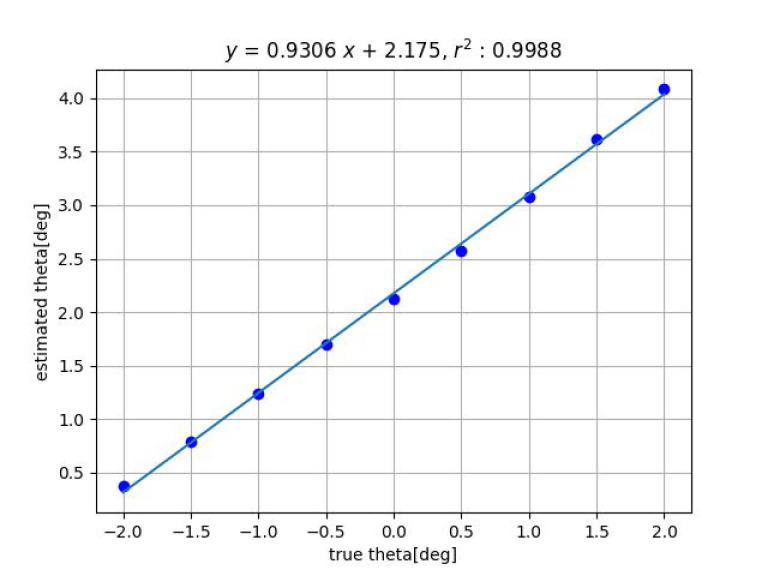
Result of tilt angle estimation by Method 1.

**Figure 10 sensors-23-00851-f010:**
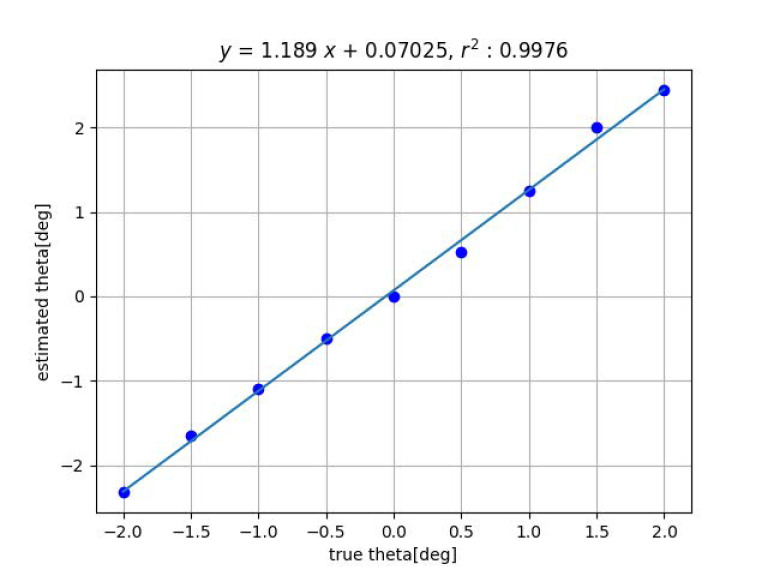
Result of tilt angle estimation by Method 2.

**Figure 11 sensors-23-00851-f011:**
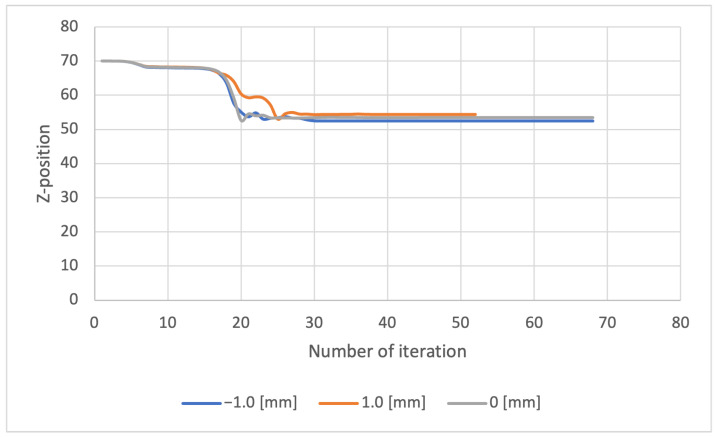
Convergence of iterative calculation of *z*-axis displacement in Method 1.

**Figure 12 sensors-23-00851-f012:**
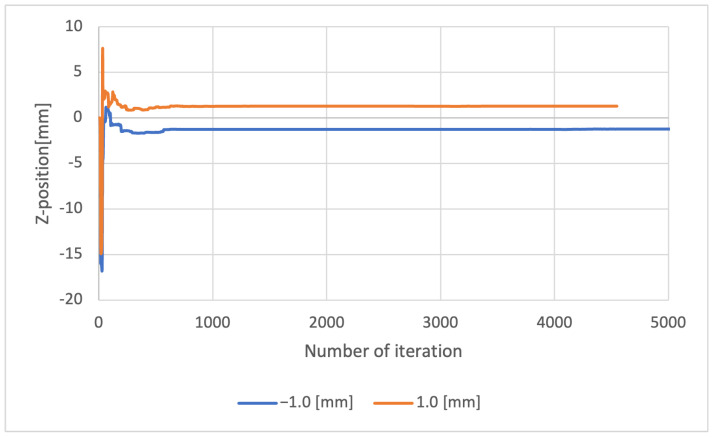
Convergence of iterative calculation of *z*-axis displacement in Method 2.

**Figure 13 sensors-23-00851-f013:**
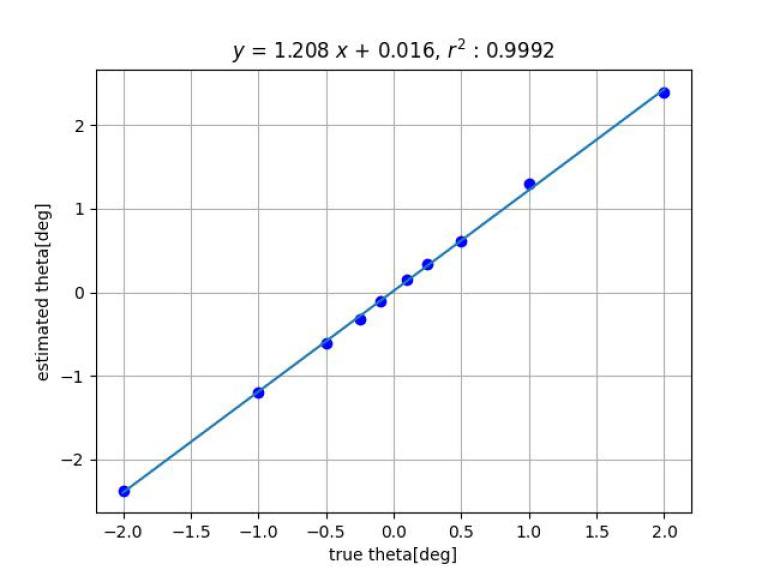
Result of estimation for tilt angle θ by method 2 without camera calibration.

**Figure 14 sensors-23-00851-f014:**
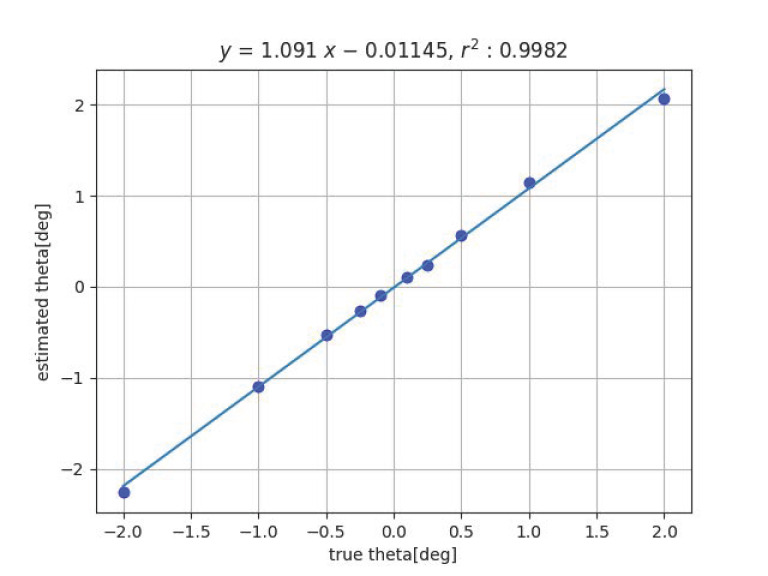
Result of estimation for tilt angle θ by method 2 with camera calibration.

**Table 1 sensors-23-00851-t001:** Summary of experimental results: The *x*-, *y*-, *z*-direction, rotation angle ψ, and tilt angle θ were each varied by the optical stage, and a straight line y=ax+b was fitted to the value *y* estimated from the image by the proposed methods for the actual amount of change *x*. $r^{2}$ represents the coefficient of determination. In addition, Each straight line was used as a test line, and $\hat{x}=\left(y-b\right)/a$ was calculated from the estimated value *y*, then the absolute maximum value of $|\hat{x}-x|$ was calculated as the maximum absolute error (max. error).

Measurement Item	Method	Slope *a*	Intercept *b*	$r^{2}$	Max. Error
displacement *x*	1	1.00	4.23	1.00	0.00550 [mm]
	2	1.14	−0.0229	0.999	0.106 [mm]
displacement *y*	1	1.06	−11.0	1.00	0.0480 [mm]
	2	1.08	0.0297	1.00	0.0289 [mm]
displacement *z*	1	0.974	53.5	1.00	0.0117 [mm]
	2	1.27	0.0317	1.00	0.0680 [mm]
rotation angle ψ	1	0.968	0.201	1.00	0.0235 [deg]
	2	0.981	0.00183	1.00	0.0264 [deg]
tilt angle θ	1	0.931	2.18	0.999	0.0696 [deg]
	2	1.19	0.0703	0.998	0.121 [deg]

## Data Availability

Not applicable.

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
