# Peer review of "Sensing Algorithm to Estimate Slight Displacement and Posture Change of Target from Monocular Images"

_sensors, 2023, doi:10.3390/s23020851_

Round 1

Reviewer 1 Report

1. In the introduction. It is recommended that the author add to the work done by other scholars in the field and state their precision.

2.  Vision technology integrated with deep learning is emerging these years in various engineering fields. The authors may add more state-of-art articles for the integrity of the introduction. For object detection, please refer to Novel visual crack width measurement based on backbone double-scale features for improved detection automation, Engineering Structures 2023. Rachis detection and three-dimensional localization of cut off point for vision-based banana robot. Computers and Electronics in Agriculture 2022.

3. There are some problems with the format of the article, for example, some line numbers are wrong.

4. In the conclusion chapter, I suggest that the author show the comparison data between the measured results of the method in this paper and the real displacement or Angle in a table for easy understanding.

5. The authors did not compare the proposed method with other advanced techniques to highlight the accuracy of the proposed method.

6. As for the test results, it is suggested that the author add steps for camera calibration, and conduct the test again after camera calibration, believing that the test results will be more accurate.

Reviewer 2 Report

The manuscript can be major revision for many reasons but these can generally be divided into technical reasons.

1. Introduction section needs revision. It should introduce some latest research results in the domain, and motivation for the proposed work.

2. The contributions of this work need to be clearly articulated. The author might consider justifying the performance of this study with recent studies and methods. In present form all looks theoretical without any justification.

3. Literature review section must also be extended. A comparative study may also be shown in graphical form

4. Lack of up-to-date references. Add/cite recent publication (2019, 2020, 2021) preferably.

5. The Limitations of the proposed study need to be discussed before conclusion.

6. Language must be improved as there are linguistic errors at some places.

7. The experiments conducted are not enough to illustrate the methodology / algorithms adopted for the work.

8. Describe clearly the abstract, conclusion and simulation part.

Round 2

Reviewer 2 Report

After reviewing all the suggestions made, I observe that they were fully carried out. Therefore, I have no more suggested changes.